# Sub-Diffraction Readout Method of High-Capacity Optical Data Storage Based on Polarization Modulation

**DOI:** 10.3390/nano14040364

**Published:** 2024-02-16

**Authors:** Li Zhang, Wenwen Li, Zhongyang Wang

**Affiliations:** 1Shanghai Advanced Research Institute, Chinese Academy of Sciences, Shanghai 201210, China; zhangli@sari.ac.cn; 2School of Microelectronics, University of Chinese Academy of Sciences, Beijing 100049, China; 3Hefei National Laboratory, University of Science and Technology of China, Hefei 230088, China

**Keywords:** optical data storage, sub-diffraction readout, polarization modulation, thin film

## Abstract

The big data era demands an efficient and permanent data storage technology with the capacity of PB to EB scale. Optical data storage (ODS) offers a good candidate for long-lifetime storage, as the developing far-field super-resolution nanoscale writing technology improves its capacity to the PB scale. However, methods to efficiently read out this intensive ODS data are still lacking. In this paper, we demonstrate a sub-diffraction readout method based on polarization modulation, which experimentally achieves the sub-diffraction readout on Disperse Red 13 thin film with a resolution of 500 nm, exceeding the diffraction limit by 1.2 times (NA = 0.5). Differing from conventional binary encoding, we propose a specific polarization encoding method that enhances the capacity of ODS by 1.5 times. In the simulation, our method provides an optical data storage readout resolution of 150 nm, potentially to 70 nm, equivalent to 1.1 PB in a DVD-sized disk. This sub-diffraction readout method has great potential as a powerful readout tool for next-generation optical data storage.

## 1. Introduction

In the past decade, we have entered an era of big data, resulting in an urgent problem of storing massive amounts of data permanently and efficiently. Since optical data storage (ODS) has an exceptional storage lifetime, it stands out among storage methods. Current optical recording in fused quartz [1,2] allows for permanent digital data storage, but the storage capacity is still limited to only 18 GB in a CD-sized disk. The present storage capacity of ODS on a GB or TB scale is largely insufficient, especially for big data centres that serve as core platforms for cloud computing, which generally operate and store data at a PB and even EB scale capacity (1 EB ≈ 10^3^ PB). Moreover, the rapid progress in Artificial Intelligence (AI) spanning scientific research and industrial development has made it necessary to store all human-generated data for AI training [3]. This pushes the need for long-term data storage to mitigate the risk of data loss. Consequently, the imminent challenge is to explore a feasible solution for high-density optical storage [4,5].

Commercial ODS, e.g., Blu-ray Disc (BD) [6,7,8], employs the confocal scanning method to achieve the recording track pitch of 320 nm and minimum recording mark length of about 150 nm with a capacity of 400 GB per disk, which is far from satisfying the exploding demand of next-generation information storage. To further improve the disk capacity, information multiplexing is exploited to extend the dimensions of ODS, i.e., wavelength [9,10], polarization [11,12,13], lifetime [14], and spatial dimensions [15,16,17,18,19]. The five-dimensional optical recording technique [20], integrated with the multiplexing of wavelength, polarization, and spatial dimensions, extends the disk capacity to 1.6 TB for a DVD-sized disk. However, the increasing of multiplexing dimensions is limited by the crosstalk between different signals, for instance, the thickness of the spacer layer should be sufficiently large to mitigate crosstalk among signals at varying depths in 3D ODS, which undoubtedly limits the layer number of the disk. Thus, the increase in dimensions only enhances the data storage capacity by several or tens of times [21,22,23]. In addition to multi-dimensional information multiplexing ODS, far-field super-resolution nanoscale writing [24,25,26] technology breaks the diffraction barrier and achieves 3D super-resolution writing, which increases the capacity of optical storage by orders of magnitude as both the lateral and axial separations are reduced. Among them, nanoscale optical writing through upconversion resonance energy transfer technology has written a lateral feature size of 54 nm, which improves the capacity of ODS to 700 TB [27], nearly at the PB scale. With further development of far-field super-resolution nanoscale writing technology, the size of recorded bits can decrease below 50 nm and the disk capacity may even exceed the PB scale [21]. Therefore, far-field super-resolution nanoscale writing technology is expected to be the solution in next-generation ODS. 

Although various methods have been exploited in far-field nanoscale writing [24,25,26,27], the efficient sub-diffraction readout method is still lacking and impedes the advancement of all-optical memory. The typical sub-diffraction readout method is fluorescence quenching microscopy (FQM) [27,28,29], which can resolve two data point spacing of 195 nm, but it restricts the lifetime of ODS due to the instability of the fluorescent medium. Another method of the sub-diffraction optical readout is reversible saturable optical fluorescence transition (RESOLFT) [30,31,32], which can resolve two data points spaced 250 nm apart [33]. Still, it decreases the readout efficiency due to the requirement for a long readout time to manipulate the fluorescence switching mechanism of the storage medium. Therefore, these methods, mainly based on the fluorescent property of the storage medium, cannot provide a practical, inexpensive, and high-efficiency ODS sub-diffraction readout.

The polarization multiplexing ODS [11,12,13,34] uses the polarizations of the writing beam to create anisotropy in recording materials and readout by polarization-dependent two-photon luminescence (TPL) [11] or absorption [12,13]. These techniques inspire us, here, to propose a sub-diffraction readout method based on such polarization modulation. Our method demodulates the written information within the diffraction limit by utilizing different polarizations of the reading beam to induce the variation in the reflected signals. To experimentally demonstrate our sub-diffraction readout method, we use a high-NA objective lens to write the information and read the written information by a low NA objective lens to achieve the sub-diffraction readout with a resolution of 500 nm, exceeding the diffraction limit by 1.2 times (NA = 0.5). In simulation to verify the capability of the sub-diffraction readout, this method can achieve a sub-diffraction readout resolution of 150 nm, potentially to 70 nm, enhancing the resolution of commercial BD by about 3 times. Combined with the multi-level polarization encoding and decoding method, the capacity of optical data storage can be improved by 1.5 times compared to the conventional binary encoding method, which enables the readout capacity of 1.1 PB in a DVD-sized disk. Moreover, in contrast to the detection of fluorescence signals, the reflected signal detection is more stable and offers a higher readout rate, which makes our method highly suitable for applications in nanoscale all-optical memory and optical encryption.

## 2. Method

### 2.1. Sub-Diffraction Readout Method

We chose a solid thin film doped with polarization-selective molecules (azo-dyes) [23] as the ODS medium. A 436 nm femtosecond writing beam with designed polarization irradiates the thin film in Figure 1a to record the information. The doped azo-dye chromophores absorb the photons and undergo *trans–cis* photoisomerization to induce the axes gradually oriented perpendicular to the polarized direction of the writing beam [35,36]. After irradiation with certain polarization directions, the absorption of the oriented molecules becomes polarization dependent anisotropy to the reading beam. Therefore, when a linearly polarized 488 nm continuous reading beam is used to irradiate the recorded point in Figure 1a, the reflectivity change depends on the angle between the polarized directions of the writing and reading beams, which is proportional to cos [2(*β* − *α*)], where *α* and *β* are the polarized directions of the writing and reading beam, respectively.

The principle of the sub-diffraction readout method based on polarization modulation is shown in Figure 1b. When we write on two adjacent data points spaced 150 nm apart with different polarizations (*β*_1_, *β*_2_), it is generally impossible to resolve the written information by confocal scanning due to the diffraction limit. To demodulate the recorded polarizations, we detect the written information with reading polarizations at *α =* 0° and *α =* 90°, respectively. Because of the linear dependence between absorption-induced reorientations with the intensity of the writing beam [35,37], the distribution of oriented molecules is proportional to the point spread function (PSF) *I_wr_*(β) of the writing beam. Thus, the reflectivity *I_α_* at different reading polarizations (α) can be described as the following:(1)$$I_{\alpha }\propto \left\{\frac{\kappa +1}{2\kappa }+\frac{\kappa -1}{2\kappa }\cos\left[2\left(\beta -\alpha \right)\right]\right\}I_{wr}*I_{rd}$$
where the ratio of the maximum (β = α) to the minimum (β = α ± 90°) of the *I*_α_ is defined as κ, which is determined by the polarization-selective feature of the materials; *I_rd_* is the PSF of the reading beam. The reflectivity of these two points at 0° and 90° reading polarizations can be described as *I*_00*°*_ = *I*_1_(*β*_1_ − 0°) + *I*_2_(*β*_2_ − 0°) and *I*_90*°*_ = *I*_1_(*β*_1_ − 90°) + *I*_2_(*β*_2_ − 90°), respectively. Therefore, the variation between *I*_00*°*_ and *I*_90*°*_ is determined by the polarizations *β*_1_ and *β*_2_ of the writing beams [38,39,40,41]. Consequently, we can demodulate the unknown recorded polarizations by comparing the variation between *I*_00*°*_ and *I*_90*°*_, as shown in Figure 1a. The variations mainly include three types: (1) peak shift, when there is a peak position shift between *I*_00*°*_ and *I*_90*°*_ while the peak values of *I*_00*°*_ and *I*_90*°*_ are almost equal, it demodulates the recorded polarizations to 0° and 90°, as shown in Figure 1c; (2) reflectivity change, when the detected peaks of *I*_00*°*_ and *I*_90*°*_ have large differences; however, without the change of peak shift and full width at half maximum (FWHM) of *I*_00*°*_ and *I*_90*°*_ (FWHMs are the same as the type (1)), it demodulates to 90° and 90°, as shown in Figure 1d; (3) FWHM change, when FWHMs of *I*_00*°*_ and *I*_90*°*_ are smaller than type (1), and the peak of the *I*_00*°*_ and *I*_90*°*_ have large differences, it demodulates to a space (nothing written) and 90°, as shown in Figure 1e. Due to the typical variation between *I*_00*°*_ and *I*_90*°*_, it can completely demodulate the recorded polarizations within the diffraction limit.

### 2.2. Polarization Encoding and Decoding Method

Based on the sub-diffraction readout method, we propose a multi-level polarization encoding and decoding method, which consists of three steps as follows:

(1) In the multi-level polarization encoding method, we encode the numbers by writing on two adjacent data points with different polarizations, and we can use *n* kinds of polarizations to write on two adjacent data points which comprise (*n* + 1)^2^ pairs of polarizations. For instance, we use 0° and 90° polarizations that comprise 9 pairs of polarizations to encode the numbers from 0 to 8, as shown in Table 1. Compared to conventional binary encoding that only stores 4 different numbers on two data points, the multi-level polarization encoding method improves the capacity of ODS by log_2_ (*n* + 1) times.

(2) In the readout process, we read the written information at 0° and 90° polarizations. When the reading polarization is 0°, the reflectivity can be given by
(2)$$I_{00{}^{\circ}}\propto \left\{\left[\frac{\kappa +1}{2\kappa }+\frac{\kappa -1}{2\kappa }cos\left(2\beta_{1}\right)\right]\right.I_{wr1}+\left.\left[\frac{\kappa +1}{2\kappa }+\frac{\kappa -1}{2\kappa }cos\left(2\beta_{2}\right)\right]I_{wr2}\right\}*I_{rd}$$
where *I_wr_* and *I_rd_* are the PSF of the writing and reading beams measured by scanning a 50 nm gold nanoparticle in confocal, as shown in Figure 1a; and *I_wr_* = 0 if nothing is written. When the reading polarization is 90°, the reflectivity can be given by
(3)$$I_{90{}^{\circ}}\propto \left\{\left[\frac{\kappa +1}{2\kappa }-\frac{\kappa -1}{2\kappa }cos\left(2\beta_{1}\right)\right]\right.I_{wr1}+\left.\left[\frac{\kappa +1}{2\kappa }-\frac{\kappa -1}{2\kappa }cos\left(2\beta_{2}\right)\right]I_{wr2}\right\}*I_{rd}$$

(3) In the polarization decoding process, due to the diffraction limit, the written information is unresolved. However, we can demodulate the recorded polarizations by comparing the variation between *I*_00*°*_ and *I*_90*°*_, and establish a relationship between the recorded polarizations and the variation between *I*_00*°*_ and *I*_90*°*_, which facilitates the demodulation into the corresponding number.

**Table 1 nanomaterials-14-00364-t001:** The Specific Encoding of 0°and 90° Polarizations.

Number	Polarizations Pair	Number	Polarizations Pair	Number	Polarizations Pair
0	a space and a space 	3	0° and a space 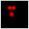	6	0° and 90° 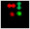
1	a space and 0° 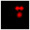	4	90° and a space 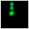	7	90° and 0° 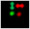
2	a space and 90° 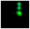	5	0° and 0° 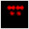	8	90° and 90° 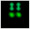

## 3. Experiment and Results

### 3.1. Polarization-Selective Feature of DR 13

To demonstrate the sub-diffraction readout method, Disperse Red 13 (DR 13) [42], one of the traditional azo-dyes, was utilized as the storage medium. Samples were prepared by spin coating on a silicon wafer with 1.5 wt% solution of poly (methyl methacrylate) (PMMA) in methylbenzene doped with 103 weight percent of DR 13 (with respect to PMMA). It was then heated on the 100 °C hot plate for 5 min to evaporate the solvent, and the thickness of the PMMA thin film was about 110 nm measured by an atomic force microscope (AFM) which is thin enough to achieve diffraction-limited writing. To demonstrate the polarization-selective feature of DR 13, we manipulated the polarizations of the 436 nm (25.5 μWcm^−2^) writing beam to write a 2 × 2 polarization array on DR 13 thin film, which is arranged in a staggered style of 90° (*t*_90°_ = 200 ms) and 0° (*t*_00°_ = 400 ms) polarization with a spacing of 1 μm, as the red arrows show in Figure 2a. Then, such a 2 × 2 polarization array is read by 0° and 90° polarization (20 nWcm^−2^), as the blue arrows show in Figure 2a. From the different reflected signals (along the yellow lines in Figure 2a) between the 0° (blue) and 90° (red) polarization reading beams, it reveals that the maximum value of the reflected signal intensity is nearly twice the minimum value in Figure 2b to provide a high contrast for the polarization modulation.

To experimentally measure the curve of cos[2(*β* − *α*)] between the reflectivity and the polarization of the reading beam, we acquired the normalized variation of the reflected signal by using a polarized reading beam with a continuous change in polarization from 0° to 180° to read the 2 × 2 polarization array Figure 2c. It found that the maximum and minimum of the reflected signal appeared when the angle between the polarizations of writing and reading beams was 0° and 90°, respectively. In Figure 2c, the DR 13 thin film exhibits a polarization-selective feature of about κ = 2. In this case, the reflected signal can be expressed by $I_{\alpha }=\left\{\frac{3}{4}+\frac{1}{4}\cos[2\left(\beta_{1}-\alpha_{1}\right)]\right\}I_{\max}$, where *I_max_* is the convolution of *I_wr_* and *I_rd_*.

### 3.2. Experiment Process

Based on the polarization-selective feature of DR 13, it remains challenging to write sub-100 nm data points with different polarizations in the thin film due to the lack of a sub-diffraction writing method. Therefore, we designed an experiment to demonstrate our sub-diffraction reading method as shown in Figure 3 The experiment includes two steps: (1) Writing process: writing different numbers on DR 13 thin film as in Figure 3a. Because of the diffraction limit, the smallest focal spot is expressed as r = 0.61 × *λ*/NA, where *λ* is the wavelength, and NA is the numerical aperture of the objective lens. We use a high NA objective lens (NA = 1.4, the smallest focal spot is about 190 nm) to write on two adjacent data points with a spacing of 500 nm. Then, we first read the written information with a high NA objective lens to confirm the accuracy of recorded polarizations Figure 3b. (2) Reading process: the written information cannot be resolved by using a low NA objective lens (NA = 0.5) directly because the diffraction limit is about 600 nm. Then, we used 0° and 90° polarizations to read the written information, and the variations of the reflected signal can be used to demodulate the corresponding numbers Figure 3c.

Writing by a High NA Objective Lens

In the writing process, we wrote the numbers from 0 to 5 on two data points with a spacing of 500 nm. Considering the different responses of DR 13 thin film to polarizations of the writing beam, we used equal intensity of writing beams with different polarizations and only changed the exposure time to achieve diffraction-limited writing. When the exposure time of the writing beams (25.5 μWcm^−2^) at 0° and 90° polarization is 300 and 75 ms respectively, we can achieve nearly diffraction-limited writing. Then, to verify the accuracy of recorded polarizations, 0° and 90° polarizations of reading beams (20 nWcm^−2^) are utilized to read the written information using a high NA objective lens, and the specifically recorded polarizations and the variations between *I*_00*°*_ and *I*_90*°*_ are shown in Figure 4a–f. To evaluate the accuracy of the writing process with different polarizations, we used Equations (2) and (3) to simulate the reflected signal variations of the written information read by 0° and 90° polarizations; the results are shown in Figure 4(a2–f2). Among the simulations, *I_wr_* and *I_rd_* are diffraction-limited PSFs of 436 nm and 488 nm, respectively. Gaussian white noise was introduced in *I_rd_* with an SNR (Signal to Noise Ratio) of 10, matched as the experiment condition. Compared to the variation of reflected signal in experiments, it reveals that the feature size of data points in the experiment is 330 nm Figure 4(d1) slightly larger than 285 nm Figure 4(d2) in our simulation. At the equal intensity of the reading beam, the maximum of the reflected signal under 0° and 90° writing polarizations are almost equal in Figure 4(b1), which is consistent with the simulation result in Figure 4(b2). The intensity varied ratio of the reflected signal *I*_00*°*_ and *I*_90*°*_ (*I*_00*°*_/*I*_90*°*_) in Figure 4(c1) is about 2.2, larger than 2 of the simulation in Figure 4(c2). However, the variation between *I*_00*°*_ and *I*_90*°*_ in the experiment, as shown in Figure 4(a1–f1), is still consistent with the simulation results in Figure 4(a2–f2). Therefore, this written information can be employed as a standard template for sub-diffraction readout with a low NA objective lens and Equations (2) and (3) can accurately describe the experimental process.

B.Reading by a Low NA Objective Lens

In the reading process, we used a reading beam (5 nWcm^−2^) with 0° and 90° polarization to read the above standard template by a low NA objective lens, and the measured reflected signals of *I*_00*°*_ and *I*_90*°*_ are shown in Figure 5a–f. To anticipate the sub-diffraction reading process, we also used Equations (2) and (3) to simulate the reflected signal variations of the standard template read by 0° and 90° polarizations using a low NA objective; the results are shown in Figure 4(a2–f2). Compared to the experimental results, the FWHM of *I*_00*°*_ in experiments (1000 nm in Figure 5(a1) and 660 nm in Figure 5(d1)) are slightly larger than those predicted in our simulation (860 nm in Figure 5(a2) and 640 nm in Figure 5(d2), because the writing size (330 nm) in the standard template is larger than the diffraction limit (285 nm). However, the primary variations in intensity change, peak shift, and FWHM change still remain. Therefore, the written information can be decoded by the primary variations in the reflected signal intensity, as shown in Figure 5, as follows: (1) intensity change, when the FWHMs of *I*_00*°*_ and *I*_90*°*_ are about 1000 nm with large intensity difference, it is decoded to 1 or 3. If *I*_00*°*_ < *I*_90*°*_, it is decoded to 1, as shown in Figure 5(a1); if *I*_00*°*_ > *I*_90*°*_, it is decoded to 3 in Figure 5(c1); (2) peak shift, when the FWHMs of the reflected signals are about 1000 nm with 120 nm peak shift less than 150 nm in the simulation, it is decoded to 2, as shown in Figure 5(b1); (3) FWHM change, when the FWHMs of the reflected signals are about 660 nm less than 1000 nm, it is decoded to 4 or 5. If *I*_00*°*_ < *I*_90*°*_, it is decoded to 4 in Figure 5d; If *I*_00*°*_ > *I*_90*°*_, it is decoded to 5 in Figure 5(e1); (4) When there is no reflected signal, it is decoded to 0 in Figure 5(f1). Therefore, our method can achieve a resolution of 500 nm, exceeding the diffraction limit by 1.2 times using an NA = 0.5 objective lens. Meanwhile, the consistency of experimental and simulation results implies that our simulation can anticipate the experiments effectively and accurately.

The sub-diffraction readout method reported here, which detected the reflected signal modulated by the reading polarization, facilitated the readout speed of ODS. By comparison, the sub-diffraction readout method based on the fluorescence intensity detection mode of RESOLFT, which irradiates each pixel for about 30 ms confined by the fluorescent switching time. Without this limitation, our method only utilizes 1 ms Figure 5a–f of irradiation for each pixel, which is confined by the scanning speed of the nanostage. It is approximately 30 times faster than RESOLFT, and considering our method needs to read twice by different polarization, it still results in a readout speed of 15 times faster than RESOLFT.

To measure the robustness and number error rate of the sub-diffraction readout method, we wrote a 5 × 5 numbers array by a high NA objective lens with a spacing of 1.5 μm, and read it by a low NA objective lens, as shown in Figure 6a. In the variations in the reflected signal, the primary difference is the peak value change in *I*_00*°*_ and *I*_90*°*_. Therefore, we distinguished between 1 and 3 by comparing the ratio of *I*_90*°*_ and *I*_00*°*_ (*I*_90*°*_/*I*_00*°*_) to 1. As shown in Figure 6b, when the FWHMs of the reflected signals are approximately 1000 nm, and the *I*_90*°*_/*I*_00*°*_ = 1.182 ± 0.032 > 1, they can be correctly decoded to 1; or the *I*_90*°*_/*I*_00*°*_ = 0.786 ± 0.033 < 1, it is decoded to 3, as shown in Figure 6b. In the same way, as shown in Figure 6c, when the FWHM of the reflected signal is approximately 660 nm, and the *I*_90*°*_/*I*_00*°*_ = 1.063 ± 0.045 > 1, it is decoded to 4; or the *I*_90*°*_/*I*_00*°*_ = 0.787 ± 0.113 < 1, it is decoded to 5, as shown in Figure 6c. The obvious difference between *I*_90*°*_/*I*_00*°*_ and 1 confirms that our method is robust. Therefore, all numbers have been demodulated correctly without any number errors, which verifies that the sub-diffraction readout method is reliable and repeatable.

C.The Potential of Our Sub-diffraction Readout Method

Due to the current lack of reliable sub-diffraction polarization writing methods, the size of the recorded data point is restricted by the diffraction limit. Once the break- through of sub-diffraction polarization writing methods such as two-beam sub-diffraction writing [27] or near-field writing [43] are made, our sub-diffraction readout method can be anticipated to rapidly apply in the nanoscale data reading. In order to show the potential, we assumed that the minimum size of the recorded data point and the distance between two adjacent data points can be 50 nm and 150 nm, respectively [43], in this case, if we use a high NA objective to read the written information, we cannot decode the recorded polarizations due to the diffraction limit of 190 nm; however, if a reading beam with polarization of 0° and 90° is utilized to read the written information, we can successfully decode the written information, as shown in Figure 7a–f, wherein Equations (2) and (3) are used to simulate, an κ value of 2 is considered based on the measured polarization-selective feature of DR 13, and the Gaussian white noise with an SNR of 10 is introduced in *I_rd_*, which is the same in our experiment. In Figure 7a–f, with the increased NA of the reading objective, the FWHMs of *I*_00*°*_ and *I*_90*°*_ decrease obviously (297 nm in Figure 7a and 180 nm in Figure 7d). However, the three primary variations between *I*_00*°*_ and *I*_90*°*_ still remain, which can be easily demodulated to the corresponding numbers. Therefore, with our experiment parameters, our sub-diffraction readout method can achieve an ODS readout resolution of 150 nm.

To further demonstrate the capability of the sub-diffraction readout method, we analyzed the important parameter of κ, which is the key to determining the resolution. Here, we introduced the number error rate to evaluate the impact of κ on the resolution, which is the minimum distance between the two adjacent data points that can be read out without any decoded number errors, as shown in Figure 7g. We found that the resolution increases with the κ values. Normally, the κ value depends on the polarization-selective feature of the storage mediums, for example, the κ of the azo-dye chromophores is 2 [43,44,45,46,47], for gold nanorods, it is nearly 5 [11,20], and for liquid crystal, it is more than 10 [48]. Therefore, for κ = 2, the resolution of the sub-diffraction method is 150 nm; when κ = 5, the resolution can achieve 90 nm, even up to 70 nm at κ = 10. Therefore, when polarization-selective material with a high κ value is chosen as the storage medium, the sub-diffraction method has the potential to achieve a resolution of 70 nm. When more than two adjacent data points are within the diffraction limit, our method still has the capability to read them out. Combining the 1.5-fold capacity improvement of the encoding method, our method has the potential to achieve a readout capacity approaching 1.1 PB in a DVD-sized disk.

## 4. Conclusions

We developed a practical sub-diffraction readout method based on the polarization modulation of polarization-selective materials. It provides an ODS readout resolution potentially to 70 nm, enhancing the resolution of commercial BD by about 3 times, which enables a readout capacity of 1.1 PB in a DVD-sized disk. In the experiments, we achieved a high accuracy sub-diffraction readout with a 500 nm resolution, exceeding the diffraction limit by 1.2 times (NA = 0.5), and prove the sub-diffraction readout method is reliable and robust by consecutively reading out a 5 × 5 numbers array without any errors. Compared with the current sub-diffraction fluorescence readout method, our method based on the detection of reflected signal intensity offers a higher stability and readout speed, which is a 15 times faster level than the RESOLFT. Moreover, compared to the FQM or RESOLFT, our sub-diffraction readout system is simple, which only needs to add a polarizer and half-wave plate in confocal microscopy. Therefore, our method provides a convenient, high-efficiency, and inexpensive way to read out the nanoscale written information, which inspires us to apply a new dimension of polarization in far-field super-resolution nanoscale writing achieving high-efficient and stable nanoscale all-optical memory. Overall, the sub-diffraction readout method of high-capacity optical data storage has application potential in nanoscale optical readout for an inexorable era of big data storage, removing the obstacles for the reading and writing integration of nanoscale optical storage.

## Figures and Tables

**Figure 1 nanomaterials-14-00364-f001:**
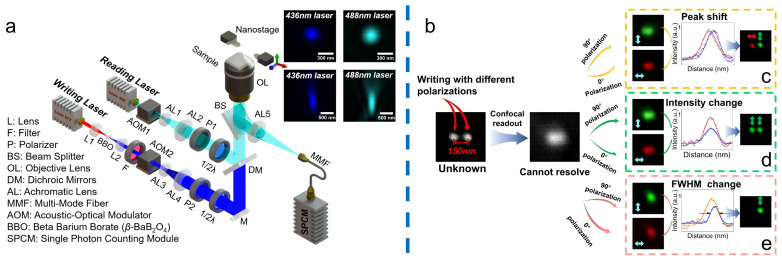
Schematic diagram of the sub-diffraction readout method. (**a**) Integrated sub-diffraction writing and reading system. The writing beam is a 436 nm femtosecond laser, which is produced by frequency doubling of an 872 nm femtosecond laser, and the reading beam is a 488 nm continuous laser. The reflectivity is detected by a single photon counting module. (**b**) Principle of the sub-diffraction readout method. Adjacent data points spaced 150 nm apart are utilized to write with different polarizations, and we demodulate the recorded polarizations by comparing the variations of the reflectivity between the reading polarizations of 0°and 90°, mainly including peak shift (**c**), intensity change (**d**), and FWHM change (**e**).

**Figure 2 nanomaterials-14-00364-f002:**
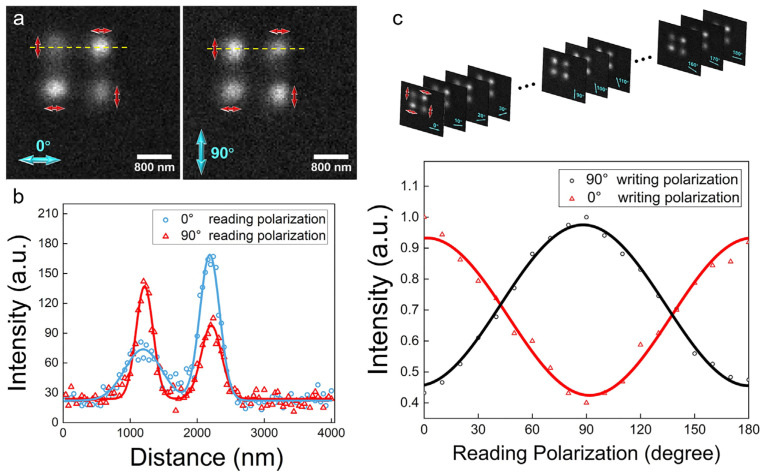
Polarization-selective feature of DR 13. (**a**) Demonstration of polarization optical data storage, the red arrow is the different polarizations of the writing beam and the blue arrow is the polarization of the reading beam. (**b**) Different reflected signal (along the yellow lines in (**a**)) between the 0° (blue) and 90° (red) polarization reading beam. (**c**) A different polarization reading beam from 0° to 180° (blue arrow) is used to read the same 2 × 2 polarizations writing array. The normalized intensity variation of the reflected signal was acquired when the writing polarization was 0° or 90° and the reading beam polarization varied continuously from 0° to 180°.

**Figure 3 nanomaterials-14-00364-f003:**
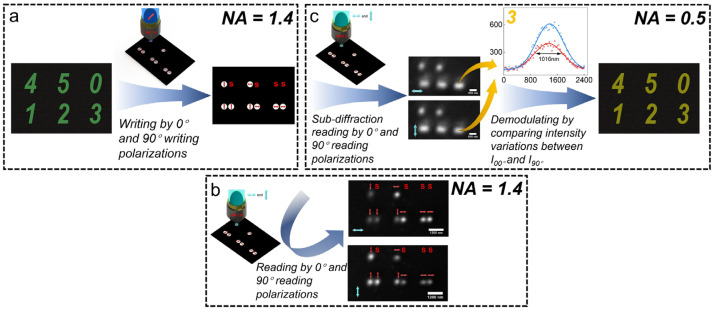
Sub-diffraction readout method of ODS. (**a**) Multi-level polarization writing by 0° and 90° polarizations using a high NA objective lens (NA = 1.4; S, space). (**b**) Reading the written information by 0° and 90° polarizations with a high NA objective lens. (**c**) Sub-diffraction reading the recording information by 0° and 90° polarizations using a low NA objective lens (NA = 0.5) and demodulating it to the corresponding number by comparing the variation of the reflected signal.

**Figure 4 nanomaterials-14-00364-f004:**
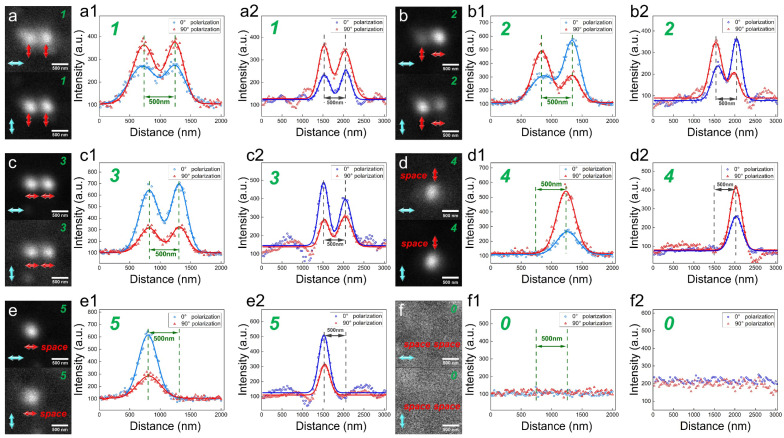
Writing the multi-level polarization encoded numbers by 0° and 90° polarization writing beam using a high NA objective. (**a**–**f**) Reading the written information by 0° and 90° polarizations (blue arrow) using a high NA objective lens. Different recorded polarizations (red arrow) on two adjacent data points 500 nm apart represent the numbers from 0 to 5 (green number); (**a1**) 90° and 90° represent 1, (**b1**) 90° and 0° represent 2, (**c1**) 0° and 0° represent 3, (**d1**) A space and 90° represent 4, (**e1**) 0° and a space represent for 5, (**f1**) A space and a space represent for 0. (**a2**–**f2**) The simulation results of the corresponding variation of *I*_00*°*_ and *I*_90*°*_ using a high NA objective.

**Figure 5 nanomaterials-14-00364-f005:**
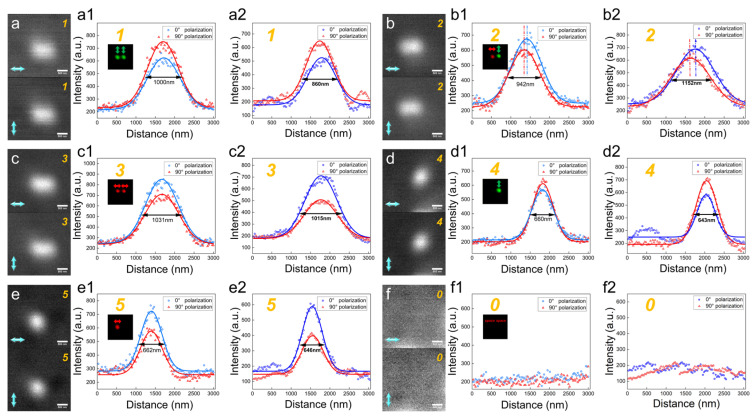
(**a**–**f**) Reading the written information by 0° and 90° polarizations using a low NA objective lens, and demodulating to the corresponding numbers from the variations in the reflected signal between 0° and 90° reading polarizations. (**a1**) *I*_00*°*_ < *I*_90*°*_, FWHMs are about 1000 nm with no peak position shift, decoded to 1 (orange number). (**b1**) Peak position shift is about 120 nm, decoded to 2. (**c1**) *I*_00*°*_ > *I*_90*°*_, FWHMs are about 1000 nm, no peak position shift, decoded to 3. (**d1**) *I*_00*°*_ < *I*_90*°*_, FWHMs are about 660 nm with no peak position shift, decoded to 4. (**e1**) *I*_00*°*_ > *I*_90*°*_, FWHMs are about 662 nm with no peak position shift, decoded to 5. (**f1**) No written information, decoded to 0. (**a2**–**f2**) The simulation of the corresponding variation between *I*_00*°*_ and *I*_90*°*_ using a low NA objective lens.

**Figure 6 nanomaterials-14-00364-f006:**
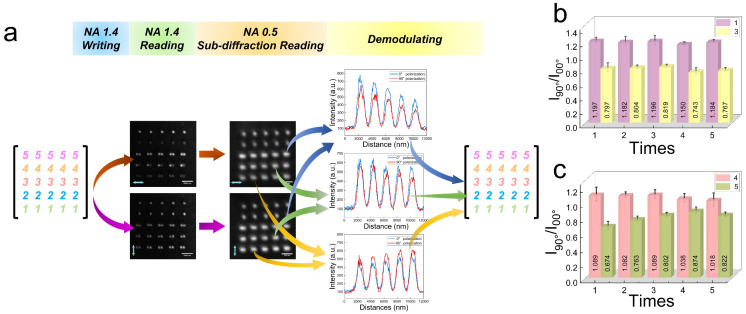
Robustness and number error rate analysis. (**a**) The process to measure the robustness of the sub-diffraction readout method. (**b**) Histogram of *I*_90*°*_/*I*_00*°*_ to distinguish between the number 1 and number 3. (**c**) Histogram of *I*_90*°*_/*I*_00*°*_ to distinguish between the number 4 and number 5.

**Figure 7 nanomaterials-14-00364-f007:**
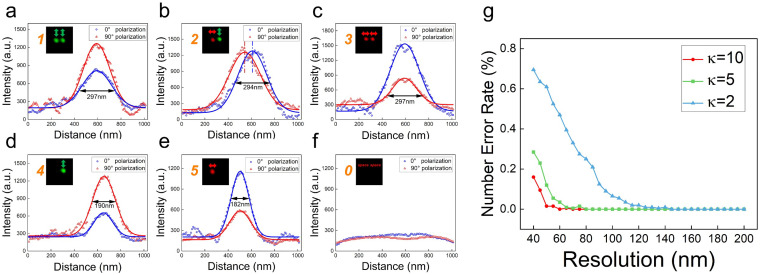
The capability of the sub-diffraction readout method to resolve two adjacent data points 150 nm apart. (**a**–**f**) The variations in the reflected signal between 0° and 90° reading polarization, and decodes the recoding information to the corresponding number from 0 to 5 (orange number). (**g**) The relationship between the number error rate and the distance between two adjacent data points. The resolution is the minimum distance between two adjacent data points that can be read out without any number errors.

## Data Availability

Data sharing is available upon reasonable request.

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
