# Peer review of "Sub-Diffraction Readout Method of High-Capacity Optical Data Storage Based on Polarization Modulation"

_nanomaterials, 2024, doi:10.3390/nano14040364_

Round 1

Reviewer 1 Report

Comments and Suggestions for Authors

This manuscript presents a novel sub-diffraction readout method using polarization modulation, achieving a resolution of 500 nm on Disperse Red 13 thin film, surpassing the diffraction limit by 1.2 times and enhancing Optical Data Storage (ODS) capacity by 1.5 times. The proposed polarization encoding method could potentially enable readout resolutions as fine as 150 nm, potentially reaching 70 nm, equivalent to a storage capacity of 1.1 petabytes in a DVD-sized disk, illustrating its potential for advanced optical data storage. While the manuscript is well-written, I have some suggestions to further improve its quality:

  1. The study primarily focuses on recording different optical information within a defined space through polarization modulation. However, the introduction lacks a comprehensive description of how such research has been conducted so far. It would be beneficial to include relevant references for a more extensive overview of related studies (e.g.,https://www.nature.com/articles/s41377-018-0095-9, https://onlinelibrary.wiley.com/doi/abs/10.1002/adfm.201908592, https://onlinelibrary.wiley.com/doi/10.1002/adma.201003374)

  2. Considering the capability of representing different optical information at a sub-diffraction level through polarization, it would be interesting to explore how this research could be extended in various ways. For instance, including comments on the characteristics that could be obtained using circular polarization would greatly enhance the paper.

  3. Some of the legends in the figures are too small, making them difficult to read. Improving their legibility would enhance the overall quality of the manuscript.

Comments on the Quality of English Language

Minor corrections are needed.

Reviewer 2 Report

Comments and Suggestions for Authors

The paper is interesting also for users who deal with other subjects so I suggest adjusting the Method section to make the formulas in lines 100-120 more readable. Otherwise, just cite the following formulas 1 and 2. Also, do not collapse Figures 1a and 1b as these appear now. Again, for readability purposes only.

It would be also of interest to the readers to understand if there is a way to turn this method, apparently write-only, into a read-write method.

Also, the authors should expand on the potential application of such dense storage. For example, is there any possible application to use these stored bits as quantum dots?

Also, does the environment temperature affect the write or read processes?

For the reviewer's curiosity: the authors say that "the minimum size of the recorded data point and the distance between two adjacent data points can be 50 nm and 150 nm". As these numbers are linked to the wavelength of the wave used in the subdiffraction process, I wonder why do not use smaller wavelengths using particles, electrons for example, instead of photons, as it is done in an electron microscope. I guess there are many reasons for not doing that but the readers - like me - would appreciate an brief explanation.
